# Thyme (*Thymus quinquecostatus* Celak) Polyphenol-Rich Extract (TPE) Alleviates HFD-Induced Liver Injury in Mice by Inactivating the TLR4/NF-κB Signaling Pathway through the Gut–Liver Axis

**DOI:** 10.3390/foods12163074

**Published:** 2023-08-16

**Authors:** Xialu Sheng, Lixia Wang, Ping Zhan, Wanying He, Honglei Tian, Jianshu Liu

**Affiliations:** 1College of Food Engineering and Nutritional Science, Shaanxi Normal University, Xi’an 710119, China; Sheng_xialu@126.com (X.S.); zhanping0993@126.com (P.Z.); thl0993@sina.com (H.T.); 2College of Life Sciences and Food Engineering, Shaanxi Xueqian Normal University, Xi’an 710061, China; 26025@snsy.edu.cn; 3Shaanxi Provincial Research Center of Functional Food Engineering Technology, Xi’an 710100, China; ljs13809194415@126.com

**Keywords:** *Thymus quinquecostatus* Celak, polyphenol, liver injury, gut–liver axis, TLR4

## Abstract

Non-alcoholic fatty liver disease (NAFLD) represents a significant and urgent global health concern. Thyme (*Thymus quinquecostatus* Celak) is a plant commonly used in cuisine and traditional medicine in Asian countries and possesses potential liver-protective properties. This study aimed to assess the hepatoprotective effects of thyme polyphenol-rich extract (TPE) on high-fat diet (HFD)-induced NAFLD and further explore possible mechanisms based on the gut–liver axis. HFD-induced liver injury in C57 mice is markedly ameliorated by TPE supplementation in a dose-dependent manner. TPE also regulates the expression of liver lipid metabolic genes (i.e., *Hmgcr*, *Srebp-1*, *Fasn*, and *Cyp7a1*), enhancing the production of SCFAs and regulating serum metabolites by modulating gut microbial dysbiosis. Furthermore, TPE enhances the intestinal barrier function and alleviates intestinal inflammation by upregulating tight junction protein expression (i.e., ZO-1 and occluding) and inactivating the intestinal TLR4/NF-κB pathway in HFD-fed mice. Consequently, gut-derived LPS translocation to the circulation was blocked, the liver TLR4/NF-κB signaling pathway was repressed, and subsequent pro-inflammatory cytokine production was restrained. Conclusively, TPE might exert anti-NAFLD effects through the gut–liver axis and has the potential to be used as a dietary supplement for the management of NAFLD.

## 1. Introduction

Non-alcoholic fatty liver disease (NAFLD) is one of the primary causes of chronic liver disease that is often associated with extrahepatic complications such as obesity, type 2 diabetes, and other metabolic syndromes [1]. Globally, NAFLD has already afflicted around 32.4% of the population [2]. The evolution of NAFLD is intricate and still inconclusive. Although several medications, such as farnesoid X receptor agonists and thyroid hormone receptor β agonists, are currently being evaluated, none have yet been approved for the treatment of NAFLD [3]. Thus, many researchers have concentrated on using plant-derived bioactive compounds as complementary medicine for the treatment of NAFLD due to their excellent effects and high security, particularly from medicinal and edible plants [4,5]. Polyphenol is an important type of plant-derived bioactive compound and a promising potential candidate for the treatment of metabolic disorders because of its wide spectrum of biological properties [6,7,8]. Certain polyphenols have the potential to modify multiple pathways, including those in the liver, gut, and brain, as well as the interconnections between these pathways, thereby improving various pathological indicators in patients with NAFLD [6,7,8,9].

Thyme (*Thymus quinquecostatus* Celak) is a traditional aromatic herb belonging to the family Lamiaceae that is widely distributed in China, Korea, and Japan [10]. It is also commonly consumed as a spice and brewed as tea in daily life [11]. Due to its condimental, nutritional, and therapeutic characteristics, thyme has been extensively utilized in both culinary and pharmaceutical industries throughout Asia [12]. Studies have shown that thyme extracts, particularly polyphenols, exhibit anti-diabetic, anti-inflammatory, antioxidant, as well as anti-tumor activities [11,13]. Furthermore, various experimental models of liver injury have demonstrated that thyme possesses hepatoprotective effects. Either ethanolic or methanolic extracts of thyme effectively suppress N-nitrosodiethylamine or aflatoxin-induced oxidative liver damage [14,15]. Thyme extract could ameliorate rat liver injury caused by carbon tetrachloride (CCl4) [16]. A laboratory model of alcoholic liver disease also demonstrated the liver-protective effects of thyme ethanolic extract [17]. However, to date, there is a lack of supporting evidence from in vivo studies that demonstrate the link between thyme polyphenol-rich extract (TPE) and the alleviation of HFD-induced liver injury.

Emerging research implies that the gut–liver axis is implicated in NAFLD pathophysiology [18,19]. NAFLD could be exacerbated or accelerated by the disturbed composition of intestinal flora [18]. Dysbiosis of intestinal flora is often accompanied by elevated lipopolysaccharides (LPS), which hinder the function of the intestinal barrier and enhances the permeability of the gut [20]. In turn, damaged intestinal barriers lead to intestinal bacteria and the translocation of byproducts, and LPS entering the systemic circulation and causing hepatotoxic effects by activating Toll-like receptor 4 (TLR4) and triggering pro-inflammatory innate immune responses [21,22]. In both human and animal studies, NAFLD exhibits dysregulated intestinal flora, higher intestinal permeability, and elevated LPS levels [23,24,25]. Thus, regulating gut microbiota is a target for NAFLD prevention and treatment.

In the present study, we aim to assess the hepatoprotective effects of dietary TPE on HFD-induced NAFLD in mice. Since phenolic phytochemicals are generally poorly absorbed, their bioavailability could be enhanced through the influence of intestinal microbiota [26]. Consequently, we focused on the gut–liver axis to explore the underlying mechanism of TPE on HFD-induced NAFLD. The findings would shed light on the potential of employing TPE as a strategy to prevent NAFLD.

## 2. Materials and Methods

### 2.1. Materials and Reagents

Thyme (*Thymus quinquecostatus* Celak) was harvested from Yulin, Shaanxi Province of China during its flowering period. The thyme leaves were dried at 40 °C, ground into powder and sifted through an 80 mesh, then vacuum-packed and stored at room temperature under ventilated and dry conditions for further use. The normal diet contained 20% proteins, 10% lipids, and 70% carbohydrates (D12450J), and the high-fat diet contained 20% proteins, 60% lipids, and 20% carbohydrates (D12492). Scutellarein, scutellarin, apigenin-7-O-glucuronide, rosmarinic acid, and SCFAs were obtained from Sigma-Aldrich, Inc. (St Louis, MO, USA). Biochemical kits for triacylglycerol (TC), triglycerides (TG), high-density lipoprotein cholesterol (HDL-C), low-density lipoprotein cholesterol (LDL-C), aspartate aminotransferase (AST), and alanine aminotransferase (ALT) were purchased from Nanjing JianCheng Bioengineering Institute (Nanjing, China). ELISA kits for the detection of lipopolysaccharide (LPS), tumor necrosis factor-alpha (TNF-α), interleukin 1β (IL-1β), and interleukin 6 (IL-6) were obtained from Shanghai Enzyme-linked Biotechnology Co., Ltd. (Shanghai, China). The primary antibodies used were TLR4 (GB15003, 1:1000), NF-κB p65 (GB12142, 1:1000), ACTIN (GB11519, 1:1000), and secondary antibodies of HRP-labeled goat anti-mouse IgG, HRP-labeled goat anti-rabbit IgG were purchased from Wuhan Servicebio Technology Co., Ltd. (Wuhan, China, Servicebio). Other chemical reagents were of analytical grade.

### 2.2. Preparation of TPE

Thyme powders were subjected to a mixture of ethanol, water, and acetic acid (60:40:1%, *v*/*v*) with a solid/liquid ratio of 1:35. The mixture was then incubated in a constant temperature oscillation incubator (Changzhou, China, Changzhou Guohua Electric Appliance Co., Ltd.) at 25 °C for 2 h followed by sonication for 1 h at 40 °C. The suspensions were filtered and centrifuged (6000× *g*, 4 °C, 10 min). Solvent was then removed from the supernatant using a rotary evaporator. DM-301 macroporous resin (Xi’an, China, Xi’an Jingbo Biotechnology Co., Ltd.) was used for the purification of TPE. The obtained extract was freeze-dried into powder and then stored at −40 °C until use.

### 2.3. Qualitative and Quantitative Analysis of the Main Components of TPE

A qualitative analysis of TPE was conducted based on previous studies [18]. UPLC-MS/MS was used to identify the constituents of brown TPE powder and HPLC was used to quantify the major components of TPE. UPLC-MS/MS analysis was performed using a Shimadzu Nexera X2 UPLC system coupled to tandem mass spectrometry (Waltham, Mass, USA, Applied Biosystems). An Agilent 1200 HPLC system (Palo Alto, CA, USA) was used. The specific procedures of TPE component analysis were described in the Appendix A.

### 2.4. Animals and Experimental Protocols

Healthy male C57BL/6J mice (6 weeks old; 18–20 g) were housed as follows: 23 ± 2 °C, 60 ± 5% relative humidity, and a 12-h cycle of light/dark. After 7 days of acclimatization, the mice were randomly divided into four groups (*n* = 8): the normal diet (ND) group, the high-fat diet (HFD) group, the HFD plus 100 mg/kg/day TPE intervention group (HFD and LTP), and the HFD plus 400 mg/kg/day TPE intervention group (HFD and HTP) for 9 weeks. Free tap water and food were provided to the mice. According to the regulations of Shaanxi Normal University’s Experimental Animal Administration, all experimental procedures were followed. This study was approved by the Committee on Care and Use of Laboratory Animals of Shaanxi Normal University, China (C31871752).

### 2.5. Oral Glucose Tolerance Test (OGTT) and Insulin Tolerance Test (ITT)

At week 8, in the OGTT, mice fasted for 12 h and then intragastrically gavaged with glucose (2.0 g/kg bodyweight). In the ITT, mice fasted for 6 h and then were injected intraperitoneally with insulin (0.75 unit/kg bodyweight). Post-administration blood glucose levels were monitored with a glucometer at 0, 30, 60, 90, and 120 min.

### 2.6. Biochemical Measurements

The serum levels of TC, TG, HDL-C, LDL-C, AST, and ALT were determined with biochemical kits. A ELISA kit was used to determine the serum levels of LPS, TNF-α, IL-1β, and IL-6.

### 2.7. Histopathologic Evaluation and Immunofluorescence (IF) Staining

Liver, epididymal adipose, and interscapular brown adipose tissues were fixed in 4% (*v*/*v*) paraformaldehyde before being embedded in paraffin. The tissue was sectioned into 4 to 10 µm sections and stained with hematoxylin-eosin (H&E). Oil-Red-O staining was performed to observe liver intracellular lipid accumulation. An optical microscope was used to observe the sections.

IF staining starts with antigen repair. Subsequently, the colon sections were incubated with primary antibodies overnight at 4 °C, then covered with the corresponding secondary antibody and kept at room temperature for 50 min without light. Cell nuclei were counterstained with DAPI staining (Wuhan, China, Servicebio) and analyzed under a fluorescence microscope (Japan, Nikon).

### 2.8. Real-Time Quantitative Polymerase Chain Reaction (PCR)

The total RNA of hepatic tissue was extracted using an RNA Extraction reagent (Wuhan, China, Servicebio) and the cDNA synthesized from quantitative RNA using a SweScript All-in-One RT SuperMix for qPCR (One-Step gDNA Remover) (Wuhan, China, Servicebio). RT-qPCR (qPCR) was performed using 2×SYBR Green qPCR Master Mix (None ROX) (Wuhan, China, Servicebio) on a CFX Connect Real-Time PCR System (Hercules, CA, USA, Bio-Rad). The relative expression of mRNA was determined by relative quantification using glyceraldehyde-3-phosphate dehydrogenase (GAPDH) as the internal reference. The primer sequences are shown in Appendix A.

### 2.9. Western Blot Analysis

The protein concentration in hepatic tissue was measured using a Bicinchoninic Acid (BCA) protein quantitative detection kit (Wuhan, China, Servicebio). Total proteins were separated by sodium dodecylsulphate polyacrylamide gel electrophoresis (SDS-PAGE). After electrophoresis, the proteins were transferred from the gel to polyvinylidene difluoride (PVDF) membranes (Wuhan, China, Servicebio). The membranes were blocked for 30 min with 5% defatted milk solution at room temperature and then incubated with primary antibodies overnight at 4 °C. Following PCR washing, the membranes were incubated for 30 min with secondary antibodies at room temperature. After washing with Tris-buffered saline containing 0.05% Tween 20 (TBST), membranes were exposed to ECL reagents (Wuhan, China, Servicebio). Finally, the blots were visualized by the chemiluminescence apparatus (Shanghai, China, CLINX).

### 2.10. Measurement of Short-Chain Fatty Acids (SCFAs)

SCFAs in the colon contents were measured in accordance with earlier laboratory methods [4]. GC-MS (Japan, Shimadzu) was used to determine the contents of SCFAs. SCFA peak identities were based on an external standard purchased from Sigma Chemical. The method of SCFA analysis is described in the Appendix A.

### 2.11. UHPLC-QTOF MS Analysis of Serum Metabolomic Analysis

The serum metabolomics analysis was adapted from Qiu et al. with slight modifications [27]. Before analysis, each serum sample was mixed with four-fold (*v*/*v*) ice-cold methanol/acetonitrile (1:1, *v*/*v*). The supernatant (0.1 mL) was obtained by centrifugation (4 °C, 14,000× *g*, 15 min) and then added to the acetonitrile/water (1:1, *v*/*v*) solvent as an internal standard.

A UHPLC system (CA, USA, Agilent Technologies) equipped with an Acquity BEH C18 column (100 mm × 2.1 mm, 1.7 um) was used for serum metabolomic analysis. Binary gradients A and B were set as 25 mM ammonium acetate and 25 mM ammonium hydroxide aqueous solution and acetonitrile, respectively. Column temperature and flow rate were set at 25 °C and 0.5 mL/min, respectively. Separation was performed at the following conditions: 0–0.5 min, 95% B; 0.5–7 min, 95–65% B; 7–8 min, 65%–40% B; 8–9 min, 40% B; 9–9.1 min, 40–95% B; 9.1–12 min, 95% B. Compound identification and relative quantitative analyses were based on the mzCloud, mzVault, and MassList databases.

### 2.12. 16s rDNA Sequencing

A cetyltrimethylammonium bromide (CTAB)/sodium dodecyl sulfate (SDS) method was used to extract DNA from the mice’s colon contents and determine the purity and concentration of the extracted DNA. PCR amplification of the bacterial 16S rRNA V3–V4 hypervariable regions was performed. Paired-end sequencing was conducted using the Illumina MiSeq platform (San Diego, CA, USA, Illumina). Quantitative Insights into Microbial Ecology (Boulder, CO, USA, QIIME, version 1.8.0) was used to process the sequencing data.

### 2.13. Statistical Analysis

Data were presented as the mean ± standard error (SEM). Statistical significance between groups was evaluated by one-way analysis of variance (ANOVA). GraphPad Prism 9.0 (San Diego, CA, USA, GraphPad Software LLC.) was used to analyze statistical differences and plot drawings. In all statistical comparisons, *p* < 0.05 was considered significant.

## 3. Results

### 3.1. Chemical Profiling of TPE

UPLC-MS/MS was used to investigate the chemical composition of TPE. In the analysis of TPE (Appendix A), a total of 46 components were detected, with flavonoids and phenolic acids emerging as the primary compounds in TPE. The HPLC analyses presented in Figure 1A,B and Appendix A indicated that the four primary polyphenols in TPE had the highest concentration. Specifically, scutellarin (231.56 ± 1.41 mg/g), apigenin 7-O-glucuronide (52.10 ± 0.79 mg/g), rosmarinic acid (206.28 ± 1.12 mg/g), and scutellarein (68.18 ± 0.55 mg/g) were identified as the predominant polyphenols in TPE.

### 3.2. Effects of TPE on Weight Parameters, Lipid Homeostasis, and Insulin Resistance of HFD-Fed Mice

To investigate the protective effects of TPE against NAFLD, HFD-fed mice were given different daily doses of TPE (100 or 400 mg/kg) orally for 9 weeks. Compared to the HFD group, TPE treatment significantly decreased body weight gain (Figure 2A,B). However, there is no discernible variation in food consumption among HFD-fed groups, indicating that the TPE effect was not attributed to a reduction in food intake (Figure 2C, *p* > 0.05). In alignment with the alterations in body weight, TPE-treated HFD-fed mice also had reduced weights of the liver, subcutaneous fat, epididymal fat, and mesenteric fat, respectively (Table 1, *p* < 0.05). As shown in Figure 2D,E, adipocyte size in the epididymal and brown tissues of the HFD group was larger than that in the ND group. The size of epididymal and brown adipocytes decreased dose-dependently in the LTP treatment and HTP treatment groups. In comparison with the ND group, the HFD group exhibited (Figure 2F–I) lipid abnormalities (*p* < 0.0001 or *p* < 0.01). It is worth noting that TPE treatment can effectively reverse the changes of TC (*p* < 0.01), TG (*p* < 0.01), and LDL-C (*p* < 0.05), but there is no significant change in HDL-C (*p* > 0.05). Meanwhile, TPE-treated HFD-fed mice had lower blood glucose levels compared to HFD-fed mice (Figure 2J,K), and the areas under the OGTT and ITT curves (ACU) showed a marked improvement (*p* < 0.05) in glucose tolerance and insulin resistance, especially for HTP supplementation. Accordingly, HTP exhibits a more pronounced impact on ameliorating glucose metabolism disorders and enhancing insulin sensitivity compared to LTP, thereby mitigating the risk factors associated with NAFLD.

### 3.3. Effects of TPE on Hepatic Lipid Disorders and Liver Damage in HFD-Fed Mice

A qRT-PCR technique was then used to determine the expression of hepatic lipid metabolic genes (Figure 3A). There was a significant increase in the hepatic mRNA expression of *Hmgcr* (*p* < 0.001), *Srebp-1* (*p* < 0.01), and *Fasn* (*p* < 0.05) in the HFD group compared with the ND group, but this increase was significantly attenuated by TPE supplementation (*p* < 0.05). Compared to the ND group, the mRNA expressions of *Pparα* (*p* < 0.05) and *Cyp7a1* (*p* < 0.05) showed an obvious reduction in the HFD group. High doses of TPE increased *Cyp7a1* mRNA expression (*p* < 0.01) in HFD-fed mice. As for the gene *Ppara*, there is no difference among HFD-fed mice. Meanwhile, a dose-dependent effect of TPE on liver mRNA expression was also observed in HFD-fed mice, indicating that HTP intervention can significantly reduce liver fatty lesions. Moreover, H&E and Oil-Red-O staining also revealed an alleviating effect of TPE on hepatic steatosis (Figure 3D). The H&E staining showed an apparent accumulation of lipids in the hepatocytes’ blurred cell borders in the HFD group. However, TPE ingestion normalized liver histomorphology changes. For HTP ingestion, the hepatic tissue showed normal cytoplasm and nucleus appearance, similar to the ND group. According to Oil-Red-O staining, the livers in the HFD group displayed severe steatosis, with large and numerous liver lipid droplets, which was improved by TPE treatment. Additionally, TPE showed protective effects against liver damage caused by HFD, as evidenced by a significant reduction in serum ALT and AST levels (Figure 3B,C, *p* < 0.0001), both of which are considered indicators of liver damage.

### 3.4. TPE Improves HFD-Induced Gut Dysbiosis

Principal coordinates analysis (PCoA) revealed significant clustering in the microbiota compositions of different groups (Figure 4A), suggesting that HFD and TPE significantly altered the bacterial composition. At the phylum level, *Firmicutes* and *Bacteroidetes* dominated gut microbiota in every group (Figure 4B). Compared to the ND group, the HFD group showed a significant increase in *Firmicutes-to-Bacteroidetes* (F/B) ratio (*p* < 0.001), while TPE treatment reversed this change (Figure 4C, *p* < 0.05).

STAMP software was used to identify the most significant microorganisms at the genus level among the ND, HFD, and HFD +HTP groups (Figure 4A,B). Results showed that the HFD group had a greater abundance of *Blautia*, *Lactobacillus*, *Romboutsia*, *Enterorhabdus*, *Lachnoclostridium*, *Tyzzerella*, *Roseburia*, and *Ruminococcaceae UCG−010* than those in the ND group. Higher levels of *Ruminiclostridium 9* and *Clostridium sensu stricto 1* were found in the ND group compared to the HFD group. However, the levels of *Romboutsia* and *Lachnoclostridium* were significantly altered by the administration of HTP. Notably, significant increases in the relative abundance of *Lactobacillus*, *Ileibacterium*, and *Dubosiella* were observed with HTP addition, while the relative abundances of *Ruminiclostridium* and *uncultured* were reduced. Collectively, with TPE administration, a healthier intestinal microbiota can be achieved.

To further illustrate the correlation between NAFLD-related traits and key microorganisms, Spearman’s correlation analyses were performed (|r| > 0.5, *p* < 0.05) (Figure 4F,G). *Lactobacillus*, *Ileibacterium*, and *Dubosiella* levels were elevated with HTP administration in HFD-fed mice, which were negatively associated with the NAFLD-related parameters. It is noteworthy that the abundance of *Blautia*, *Romboutsia*, *Enterorhabdus*, *Lachnoclostridium*, *Tyzzerella*, *Roseburia*, and *Ruminococcaceae UCG−010* were significantly increased in mice exposed to HFD, and these genera were found to be positively correlated with most of the parameters related to NAFLD.

### 3.5. Effect of TPE on SCFAs

As a major class of metabolites generated in the intestine, SCFAs play a crucial role in intestinal homeostasis and NAFLD [28]. The levels of SCFAs in the colon are depicted in Figure 5. The HFD group displayed significantly lower levels of total SCFAs but this was reversed by TPE administration. Upon TPE treatment, acetic, propionic, and butyric acid production was dose-dependently upregulated. Notably, acetic (Figure 5B, *p* < 0.001) and butyric acid (Figure 5D, *p* < 0.05) were markedly elevated under the intervention of 400 mg/kg/day TPE in HFD-fed mice.

### 3.6. Effect of TPE on Microbial-Derived Serum Metabolites in HFD-Fed Mice

In metabolic disease pathogenesis, microbiota-derived metabolites play a key role [29]. A serum untargeted metabolomics analysis was performed to identify the response of metabolites to the TPE-altered gut microbiome and thereby improve NAFLD. According to PLS-DA scoring plots, there was a distinct classification for each treatment group (Figure 6A,C). A closer distance was observed between mice of the HFD + HTP group and the ND group, indicating that HTP might partially reverse the alteration in metabolites caused by HFD feeding. OPLS-DA was used to detect the difference between the HFD group and the HFD + HTP group, and the score plots (Figure 6B,D) proved a clear separation between them. Furthermore, 32 differential metabolites were identified as potential biomarkers between the HFD and HFD + HTP groups (VIP > 1.0 and *p* < 0.05). Specifically, in the HFD + HTP group, 19 metabolites were increased and 13 metabolites were decreased when compared to the HFD group (Figure 4E). The metabolic pathways activated by HTP in HFD-fed mice were identified using MetabolAnalyst 5.0. In comparison with the HFD group, TPE intervention significantly altered nine metabolic pathways (Figure 6F), including linoleic acid (LA) metabolism, glycerophospholipid metabolism, arachidonic acid (AA) metabolism, purine metabolism, sphingolipid metabolism, and so forth.

The relationship between serum differential metabolites and the key microbial organisms was illustrated in Appendix A. Several species of Ileibacterium, Dubosiella, Ruminiclostridium, uncultured Tyzzerella, Ruminococcaceae UCG-010, Lachnoclostridium, Lactobacillus, and Roseburia had significant correlations with several typical serum metabolites. Specifically, the serum metabolites LA and AA were positively correlated with Lactobacillus, which was markedly increased in the HFD + HTP group.

### 3.7. Effect of TPE on Intestinal Barrier Function and the Intestinal TLR4/NF-κB Signaling Pathway

HFD may compromise intestinal epithelial integrity, impair intestinal permeability, and cause potentially harmful antigens to enter the bloodstream. Through a continuous HFD, the intestinal barrier is damaged, as indicated by a substantial reduction in Zonula occludens-1 (ZO-1) and occluding expression (Figure 7A). However, TPE treatment was found effective in restoring the expression of ZO-1 (*p* < 0.01) and occluding (*p* < 0.01) with high doses. Moreover, no significant difference was observed between the HFD and the HFD + LTP groups.

As shown in Figure 7C, the TLR4/NF-κB signaling pathway was activated in the HFD group, manifested as the increased aggregation of TLR4, MyD88, and NF-κB, while TPE reversed this trend and restored the levels of these proteins to those seen in the ND group (Figure 7D).

### 3.8. Effect of TPE on Inflammation and the Hepatic TLR4/NF-κB Signaling Pathway

Translocation of LPS is possible when the intestinal barrier is damaged. As shown in Figure 8A, HFD feeding caused a marked increase in serum LPS compared to ND-fed mice (*p* < 0.001), but this was remarkably reduced by TPE treatment (*p* < 0.01). Moreover, systemic inflammation induced by HFD was abolished with TPE supplementation by suppressing the levels of serum IL-1β, IL-6, and TNF-α (Figure 8B–D, *p* < 0.05).

We further assessed the hepatic TLR4/NF-κB signaling pathway (Figure 8E,F) using Western blotting. The protein expressions of TLR4 (*p* < 0.05) and downstream NF-κB p65 (*p* < 0.01) were dramatically increased in the HFD group than in the ND group. Conversely, HFD-fed mice receiving TPE showed remarkable downregulation and HTP had a more pronounced inhibitory effect.

## 4. Discussion

Globally, NAFLD has emerged as a significant public health concern, underscoring the need to develop effective and safe therapeutic agents. Previous studies have confirmed that thyme exhibits notable anti-diabetic, anti-inflammatory, and antioxidant properties, and several studies have also indicated that thyme extracts may have hepatoprotective properties [12,17,18]. All these investigations demonstrated that TPE has great potential in preventing NAFLD.

In this study, qualitative and quantitative characterization of TPE was presented, as well as the compounds it contained (Appendix A & Figure 1). Over 9 weeks of treatment, TPE showed protective effects on HFD-induced NAFLD in mice, and HTP exhibited a better protective effect than LTP. According to our results, TPE intervention resulted in a significant reduction in body weight gain caused by HFD, as well as normalization of serum biochemical profiles, such as serum TC, TG, LDL-c, AST, and ALT, which were indicative of NAFLD [30]. In addition, the supplementation of TPE normalized glucose tolerance and insulin sensitivity in HFD-fed mice. Additionally, TPE inhibited mRNA expression in adipogenesis-related genes of the liver, including *Srebp-1* and *Fasn*. Meanwhile, with the intervention of TPE, hepatic *Hmgcr* was downregulated and hepatic *Cyp7a1* was upregulated, both of which reduced hepatic cholesterol levels. Hepatic and adipose morphology also supported the benefits of TPE (Figure 2D,E and Figure 3D).

The gut microbiome has been identified as a significant factor in obesity-related diseases including dyslipidemia and NAFLD [19,30]. We hypothesize that the modification of gut microbiota is primarily responsible for the benefits of TPE since phenolic phytochemicals generally possess low bioavailability and are and metabolized by the gut microbiome [26]. Consistent with previous studies, we found that HFD significantly influenced the composition of gut microbiota by increasing or decreasing the relative abundance of bacteria associated with NAFLD [4]. In detail, the *F/B* ratio that was markedly elevated in HFD-fed mice, which is associated with obesity-related diseases like NAFLD [31], was reversed by TPE treatment. Moreover, HFD-fed mice showed lower levels of *Lactobacillus*, *Ileibacterium*, and *Dubosiella* genus, while TPE administration greatly increased the abundance of these genera. Out of these, *Lactobacillus*, a kind of probiotic bacteria, has been found to have negative associations with obesity, hypertension, dyslipidemia, and NAFLD [32], as well as beneficial effects in attenuating multi-organ inflammation [33] and reducing cholesterol levels [34]. In addition, *Lactobacillus* is the most important genus for generating lactic acid and lowering intestinal PH, which encourages the synthesis of SCFAs and maintains gut health [35]. *Ileibacterium* has been reported to protect mice from adiposity [36]. Wang et al. also discovered that *Ileibacterium* may alter host metabolism and improve inflammatory disorders [37]. *Dubosiella*, involved in the synthesis of SCFAs, has also been reported to modulate metabolism, enhance intestinal immunity, and improve diseases caused by inflammation [38,39]. Furthermore, Spearman’s correlation analysis revealed negative correlations between beneficial gut bacteria (including *Lactobacillus*, *Ileibacterium*, and *Dubosiella*) and the NAFLD-related parameters. Additionally, in comparison with the HFD group, the abundance of *Romboutsia*, *Lachnoclostridium*, and *Ruminiclostridium* was decreased in the HTP-treated HFD-fed mice, whereas they have been demonstrated to be elevated in HFD-induced obese mice [40,41,42]. Meanwhile, all of them exhibited strong positive associations with the NAFLD-related parameters. In brief, through TPE intervention, the gut microbiome can be modulated toward a healthier state.

SCFAs are one of the health-beneficial metabolites of intestinal flora, which could maintain intestinal homeostasis and suppress inflammation [28,43]. As the key source of energy for intestinal epithelial cells, SCFAs, particularly butyrate, may upregulate the expression of tight junction proteins to maintain the intestinal barrier function [44,45]. In addition, SCFAs can inhibit macrophage infiltration and alleviate intestinal inflammation by reducing the formation of T-regulatory cells [46]. Moreover, SCFAs can be transported to fat, liver, and other tissues via the portal vein, promoting steatolysis by interacting with receptors [47]. In our study, TPE-treated mice had higher levels of total SCFAs, acetic acid, propionic acid, and butyric acid compared to HFD-fed mice (Figure 5). Meanwhile dose-dependent effects of TPE on SCFA levels were also observed.

Increasing evidence indicates that the host metabolome is altered by the gut flora, which in turn impacts systemic metabolism [48]. In this study, serum metabolomics analysis was conducted to elucidate the mechanisms underlying the improvement of NAFLD through TPE intervention. According to the metabolic pathway analysis, AA metabolism, glycerophospholipid metabolism, and LA metabolism are the three most important metabolic pathways (*p* < 0.05, impact > 0.1). LA is an essential fatty acid that improves lipid metabolism by lowering blood TC and LDL-C levels [49]. Our results showed that HTP enhanced LA levels, which were decreased in HFD-fed mice. Numerous metabolic disorders, including obesity and insulin resistance, have been associated with glycerophospholipid metabolism [50]. When glycerophospholipid metabolism is impaired, liver metabolism will negatively affect the metabolism [51]. Phosphatidylcholine (PC) and phosphatidylethanolamine (PE) are the most abundant phospholipids in animal cellular membranes. The PC/PE ratio in NAFLD patients was decreased, resulting in impairment of cell membrane permeability and hepatocyte dysfunction [52]. Although TPE administration did not result in significant alterations in PE concentrations in HFD-fed mice, it could improve lipogenesis in NAFLD by increasing the PC/PE ratio. There is a strong link between AA metabolism and NAFLD incidence and progression. Anti-inflammatory and antioxidative properties of AA could prevent liver damage [53]. Our findings indicated that HTP intervention increased the level of AA. Moreover, positive correlations were also found between beneficial serum metabolomics (such as LA, phosphatidylcholine, and AA) and beneficial gut bacteria (such as *Lactobacillus*, *Ileibacterium*, and *Dubosiella*).

The intestinal barrier, playing a critical role as a structural component within the gut–liver axis, functions as a vital physical and functional barrier, effectively impeding the translocation of potentially harmful luminal antigens into the circulation [54]. Clinical studies have shown that patients with NAFLD have damaged intestinal barriers [55]. It is known that TLR4 is a receptor of LPS expressed on the membranes of intestinal epithelial cells, hepatocytes, immune cells, and so on. The damaged intestinal barrier allows more LPS into the circulation and activates the TLR4 inflammatory signal pathway in intestinal epithelial cells, which negatively impacts the intestinal epithelial immune barrier and drives NAFLD progression [47]. There is growing evidence that inhibiting TLR4 could improve inflammatory conditions and enhance intestinal barrier function [56,57]. In our study, IF analysis showed that HTP improved HFD-induced intestinal pathological damage and increased intestinal permeability by increasing tight junction protein expression, i.e., ZO-1 and occluding (Figure 7A), both of which have a significant impact on tight junction integrity [58]. A significant elevation in TLR4, MyD88, and NF-κB p65 protein expression in HFD-fed mice was observed, and TPE treatments could reverse these increasing trends. Drawing upon this evidence, we propose that TPE enhances intestinal barrier function and mitigates intestinal inflammation by upregulating the expression of tight junction proteins and suppressing the TLR4/NF-κB signaling pathway, thereby restraining harmful bacterial and gut-derived LPS leakage through the intestinal barrier.

Under normal physiological conditions, only a few LPS pass through the intestinal barrier and finally reach the liver. However, HFD-induced NAFLD mice possess gut microbial dysbiosis and damaged intestinal barrier function, which both synergistically result in a large amount LPS being produced and transported into the circulation, then entering the liver. This would trigger an inflammatory response in the liver by activating the TLR4/NF-κB signaling pathway and promoting the translocation of p65 to the nucleus, which is responsible for the release of inflammatory mediator such as IL-6, ultimately leading to NAFLD [22]. Our study showed that TPE reduced concentrations of serum LPS, TNF-α, IL-6, and IL-1β and downregulated the hepatic TLR4/NF-κB signaling pathway.

## 5. Conclusions

Our study has provided evidence supporting the beneficial impact of thyme (Thymus quinquecostatus Celak)-derived TPE on HFD-induced NAFLD in mice, which had not been previously reported. In the mice models, TPE treatment effectively prevents hepatic steatosis, rectifies HFD-induced gut dysbiosis, boosts SCFA generation, and regulates serum metabolites. Furthermore, TPE enhances intestinal barrier function and ameliorates intestinal inflammation by inactivating the TLR4/NF-κB signaling pathway, thereby blocking gut-derived LPS translocation into the circulation and repressing the liver TLR4/NF-κB signaling pathway to improve liver damage. Overall, our findings imply that TPE exhibits promising potential as a dietary supplement for the prevention of NAFLD in experimental models. Moving forward, further in-depth research is warranted to explore the underlying mechanisms of TPE and its potential applications in clinical practice.

## Figures and Tables

**Figure 1 foods-12-03074-f001:**
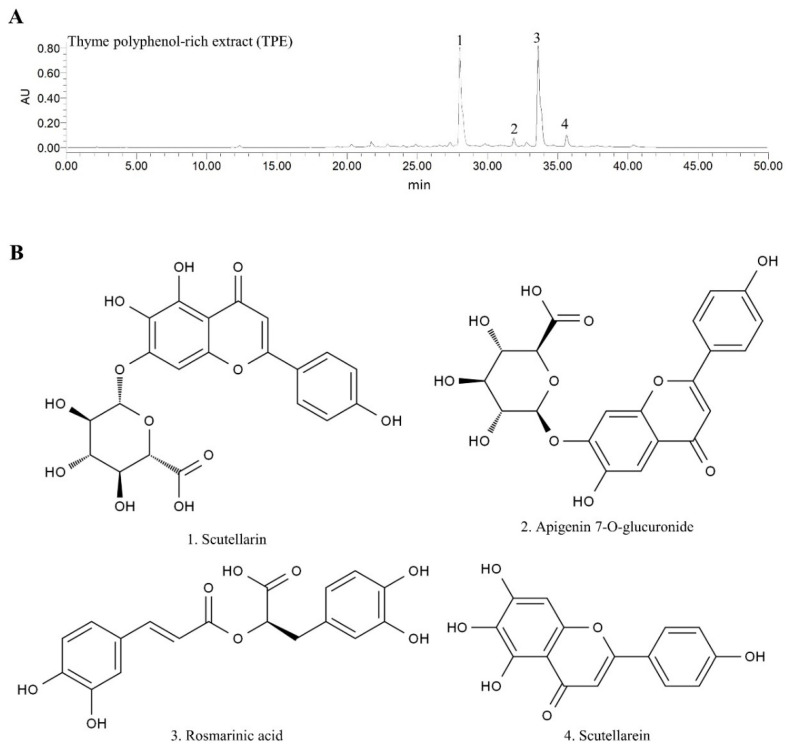
(**A**) Typical high-performance liquid chromatography (HPLC) chromatograms of Thyme Polyphenol Extract. AU, absorbance unit. (**B**) Chemical structures of the four major TPEs.

**Figure 2 foods-12-03074-f002:**
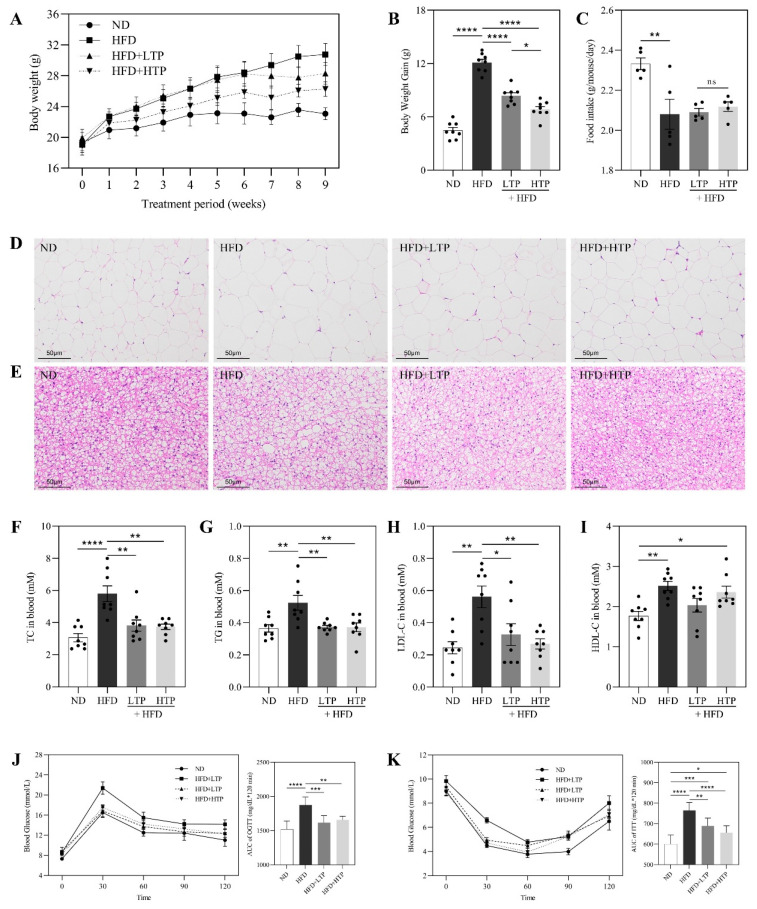
Effects of TPE treatment on the body weight, food intake, fat accumulation, and insulin resistance of mice. (**A**) Body weight of ND- and HFD-fed mice treated daily with LTP (TPE, 100 mg/kg per day) or HTP (TPE, 400 mg/kg per day) for 9 weeks. (**B**) Body weight gain. (**C**) Average daily food intake for the above four groups of mice. Representative pictures of H&E-stained (**D**) epididymal adipose tissue (200×), (**E**) brown adipose tissue (200×). Serum (**F**) triacylglycerol (TC), (**G**) triacylglycerol (TC), (**H**) low-density lipoprotein cholesterol (LDL-C), and (**I**) high-density lipoprotein cholesterol (HDL-C). (**J**) Effect of TPE on glucose tolerance measured by oral glucose tolerance test (OGTT). Right: Area under the curve (AUC). (**K**) Effect of TPE on percentage of initial blood glucose level during insulin tolerance test (ITT). Right: AUC. The data are expressed as the means ± SEM (*n* = 6–8). * *p* < 0.05, ** *p* < 0.01, *** *p* < 0.001 and **** *p* < 0.0001, indicating significant differences between these groups.

**Figure 3 foods-12-03074-f003:**
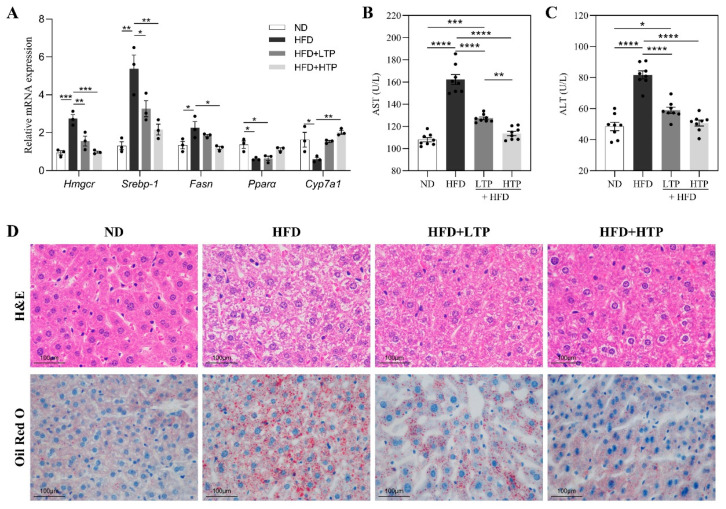
Effects of TPE on hepatic steatosis and liver damage in mice. (**A**) Hepatic lipid metabolic gene mRNA expression. Serum (**B**) aspartate aminotransferase (AST), (**C**) alanine aminotransferase (ALT). (**D**) H&E and Oil-Red-O staining of liver tissues (400×). The data are expressed as the means ± SEM (*n* = 3–8). * *p* < 0.05, ** *p* < 0.01, *** *p* < 0.001 and **** *p* < 0.0001, indicating significant differences between these groups.

**Figure 4 foods-12-03074-f004:**
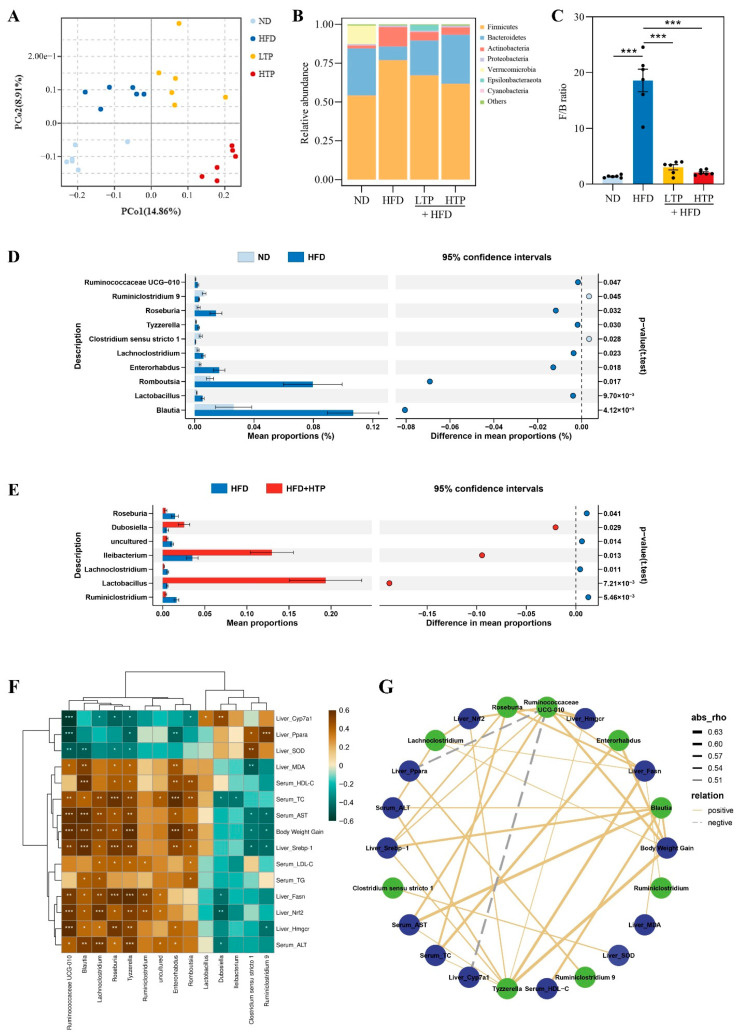
TPE modified composition of colon microbiota in HFD-fed mice. (**A**) principal coordinate analysis (PCoA) based on the OTUs level. (**B**) Phylum-level composition of colon microbiome. (**C**) The ratio of F/B. Mean proportion and their differences of the fecal microbiota genus in different group mice were analyzed by STAMP: (**D**) ND vs. HFD; (**E**) HFD vs. HFD + HTP; (**F**) heatmap of Spearman’s correlation. The intensity of the color represents the degree of Spearman’s rank correlation coefficient. (**G**) co-occurrence network. Note: *p* < 0.05, absolute value of Spearman’s rank correlation coefficient >0.5; green nodes, the gut microbial genera; blue nodes, the NAFLD-related metabolic parameters. Results are expressed as the mean ± SEM (*n* = 8). * *p* < 0.05, ** *p* < 0.01 and *** *p* < 0.001, indicating significant differences between these groups.

**Figure 5 foods-12-03074-f005:**
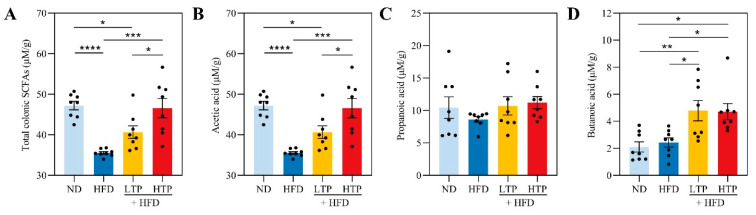
TPE altered the metabolism of the colon SCFAs in mice fed HFD. Concentrations of (**A**) total SCFAs, (**B**) acetic acid, (**C**) propionic acid, and (**D**) butyric acid. The bacterial groups and SCFAs are indicated by blue arrows and red arrows, respectively. Data are displayed as the means ± SEM (*n* = 8). * *p* < 0.05, ** *p* < 0.01, *** *p* < 0.001 and **** *p* < 0.0001.

**Figure 6 foods-12-03074-f006:**
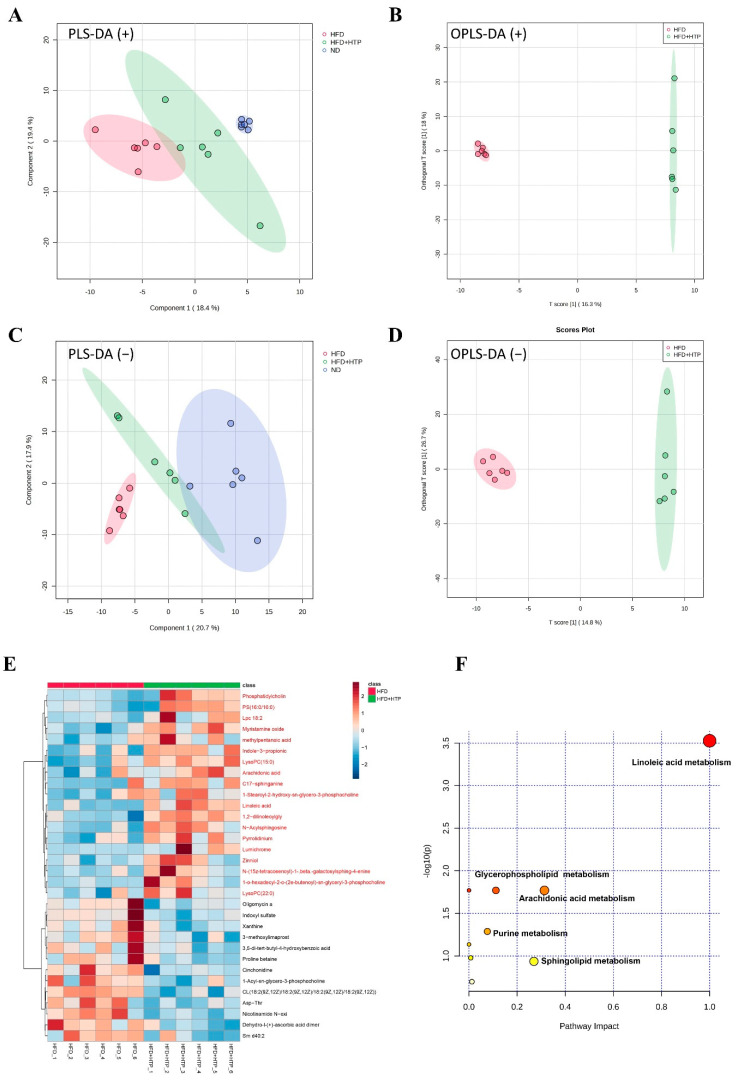
TPE modulated the serum metabonomic in mice fed HFD. Serum metabonomic profiling by UHPLC-MS/MS (ESI+). (**A**) PLS-DA score plot of serum metabolic profiles of the ND, HFD, and HFD + HTP groups; (**B**) OPLS-DA score plot of serum metabolic profiles of the HFD and HFD + HTP groups. Serum metabonomic profiling by UHPLC-MS/MS (ESI−). (**C**) PLS-DA score plot of serum metabolic profiles of the ND, HFD, and HFD + HTP groups; (**D**) OPLS-DA score plot of serum metabolic profiles of the HFD and HFD + HTP groups. (**E**) Heatmap of relative abundance of significant different metabolites (VIP > 1.0, and *p* < 0.05) in the HFD and HFD + HTP groups. (**F**) Summary of metabolic pathway analysis with MetaboAnalyst 5.0 (each point represents one metabolic pathway; the size of dot and shades of color are positive correlations with the impact of the metabolic pathway).

**Figure 7 foods-12-03074-f007:**
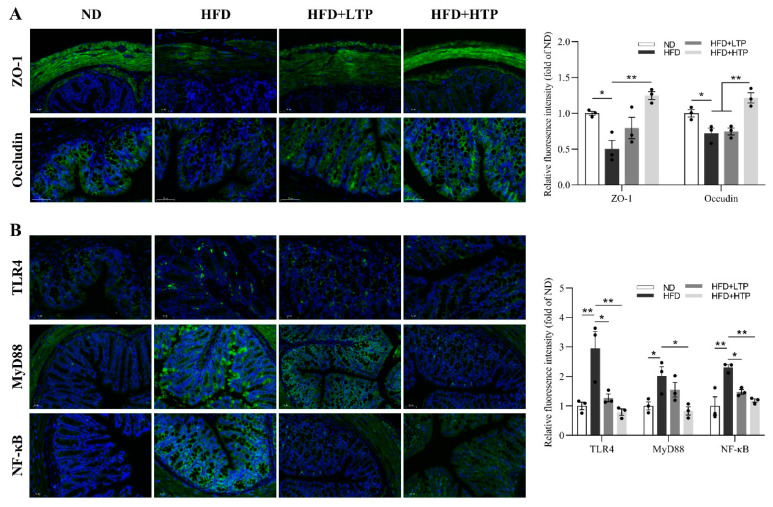
Effects of TPE on intestinal barrier function and the TLR4/NF-ΚB signal pathway in colon tissue of HFD-fed mice. (**A**) Representative images of immunofluorescence of ZO-1 and Occludin. Right: Relative fluorescence intensity of ZO-1 and Occludin. (**B**) Representative images of immunofluorescence of TLR4, MyD88, and NF-ΚB in the colon. Right: Relative fluorescence intensity of TLR4, MyD88, and NF-ΚB. The data are expressed as the means ± SEM (*n* = 3). * *p* < 0.05 and ** *p* < 0.01, indicating significant differences between these groups.

**Figure 8 foods-12-03074-f008:**
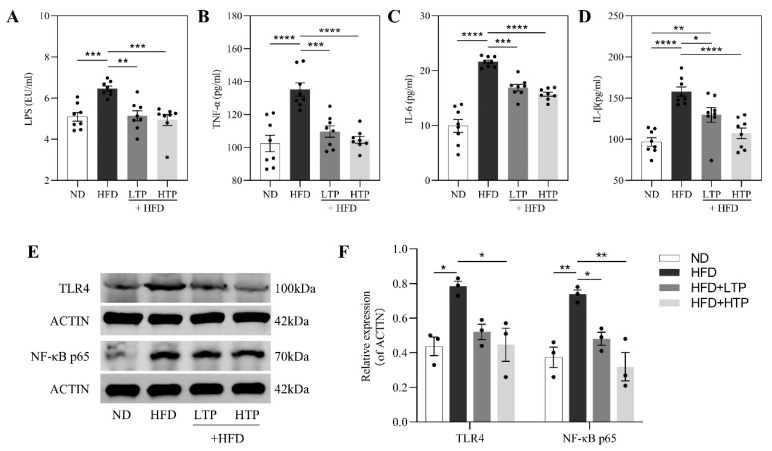
Effects of TPE on serum LPS, inflammation, and the hepatic TLR4/NF-ΚB signal pathway in HFD-fed mice. Levels of serum (**A**) LPS, (**B**) TNF-α, (**C**) IL-6, (**D**) IL-1β. (**E**,**F**) Protein expression of hepatic TLR4 and NF-ΚB p65. The data are expressed as the means ± SEM (*n* = 3–8). * *p* < 0.05, ** *p* < 0.01, *** *p* < 0.001 and **** *p* < 0.0001, indicating significant differences between these groups.

**Table 1 foods-12-03074-t001:** Supplementation of TPE on the organ indexes of mice.

	ND	HFD	HFD + LTP	HFD + HTP
liver wt (g)	0.78 ± 0.02 ^c^	1.04 ± 0.08 ^a^	0.95 ± 0.09 ^ab^	0.94 ± 0.05 ^b^
epididymal fat wt (g)	0.42 ± 0.08 ^c^	1.10 ± 0.21 ^a^	0.61 ± 0.13 ^bc^	0.66 ± 0.12 ^b^
mesenteric fat wt (g)	0.28 ± 0.03 ^c^	0.59 ± 0.09 ^a^	0.4 ± 0.04 ^b^	0.4 ± 0.04 ^b^
subcutaneous fat wt (g)	0.26 ± 0.07 ^c^	0.84 ± 0.18 ^a^	0.51 ± 0.09 ^b^	0.61 ± 0.13 ^b^
epididymal fat/body weight (%)	1.82 ± 0.35 ^b^	3.55 ± 0.71 ^a^	2.33 ± 0.48 ^b^	2.29 ± 0.59 ^b^

Data are expressed as means ± SEM (*n* = 8) and different letters in the same row indicated significant differences at *p* < 0.05. wt: weight, ND: a normal diet group, HFD: high-fat diet group, HFD + LTP: HFD plus 100 mg/kg/day TPE intervention group, HFD + HTTP: HFD plus 400 mg/kg/day TPE intervention group.

## Data Availability

The data used to support the findings of this study can be made available by the corresponding author upon request.

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
