# Peer review of "Thyme (*Thymus quinquecostatus* Celak) Polyphenol-Rich Extract (TPE) Alleviates HFD-Induced Liver Injury in Mice by Inactivating the TLR4/NF-κB Signaling Pathway through the Gut–Liver Axis"

_foods, 2023, doi:10.3390/foods12163074_

Round 1
Reviewer 1 Report
Here, the authors studied potential hepatoprotective effects of thyme polyphenol-rich extract (TPE) in the context of high-fat diet (HFD)-induced NAFLD in mice.
Major comments:
1. Abstract: Avoid the use of not defined terms (e.g. PLS, TLR4, NF-κB, etc.). Moreover, gene names have to be written in italics (see lines 19-20).
2. Methods: Sometimes part of the provider information is missing for different materials and equipment and use a uniform style for provider (city, state, county, as done for “Illumina MiSeq platform (Illumina, San Diego, CA, USA)” l. 170) information for ALL used materials and methods (e.g. see “ethanol, water, and acetic acid” in l. 84, “oscillation incubator” in l. 86, “DM-301” line 88, “UPLC-MS/MS” l. 92, “HPLC” in l. 93, “ELISA kit … LPS … TNF-α…” in l. 115-117, also see lines 120-121, 124, 125, 131, 133, 143, 149 etc.). Please also provide provider information and versions for all utilized software (e.g. see l. 171 or 173).
3. Methods (l. 100 ff): Please define the composition of the high-fat diet utilized in this study.
4. Methods (“2.11. Statistical analysis”): It is required to indicate which statistics were used for analyses (e.g. t-Test etc.).
5. Tables and figures: Please define all abbreviations used in the tables, figures and figure legends (e.g. see “AU” in l. 184, “wt” in l . 210, “ND”, “AUC” in line 213, “AST, ALT” in line 244 and so on). Use a uniform style for SI units (e.g. “l” vs “L”).
6. Figure 2: In C statistical significance should also be checked and indicated for samples “LPT and HPT”. In D and E: Please include size bars to allow the reader to evaluate the presented data (as done in Fig. 7).
7. Figure 3D: Please include size bars to allow the reader to evaluate the presented data. A vs B and C: Please use a uniform style for data presentation (bars with or without singled data points). The same applies for Fig. 4C vs Fig. 5 and Fig. 8 A-D vs F.
8. Figure 8E: Please include MW for the analysed proteins.
9. “Conclusion (473 ff): Here the authors stated “In line with our findings, we … had not been previously reported”. This seems quite optimistic and the general meaning was not shown in the present study. Here, these effects were only analysed in an artificial mouse model. Thus, the statement have to be corrected in this context (e.g. with “in mice”). The same applies for the sentences in lines 479-482.
10. “References”: Please check and correct all included references and use a uniform style. E.g. avoid the use of different types of hyphen (pages in ref. 1 vs 2; lines 519 vs 522), the use of “_” in l. 523, alternating between full vs abbreviated journal names (see/compare l. 523 vs 526) etc.
Minor comments:
a. In general, English style, phrasing, and grammar have to be revised carefully. Terms like “in vivo” have to be written in italics (e.g. see l. 58). Use a uniform style for unit, such as “min” vs “minutes” vs “hours” and so on (e.g. see lines 86, 87, 107, 110, 125, 141, 143, 152, 156 etc.). Please use a uniform style for SI units (e.g. see “µl” vs “mL” in lines 155 vs 162 etc.). Use a uniform style for significant differences (e.g. compare l. 366 “p < 0.05” vs l. 359 “P < 0.01”. Check the whole text. Please use a uniform style for the term “figure” vs “Fig.” (instead of alternating between both forms; see lines 194, 197, 200, 257 etc.). Sometimes, thousands separators are missing in numbers (e.g. see l. 87: “6000”, l. 156 etc.). Define all used abbreviations when used for the first time in the text. After the term was defined, please uniformly use the abbreviation instead of alternating between full names and the abbreviation (e.g. see: "TLR4” in l. 67, “bw” in l. 108, 109 ,” PCR” l. 128, “TBST” l. 143, “SDS-PAGE, BCA, PVDF” lines 138-140, “CTAB/SDS” in l. 167 etc.). Sometimes space characters are missing or too much (e.g. see l. 155 “100μL” vs 162 “0.5 mL/min”, l. 226 “P <0.01,” vs l. 228 “P < 0.05” vs l. 247 “p < 0.001”, also see l. 160 “25mM” etc.). In line 158, the term “ACQUITY UPLC BEH 158 C18” should not be written in capital letters. In general, the text was written in a rather sloppy style. It is recommended to consult a native English speaker and to make use or editorial services offered by the journal.
In general, English style, phrasing, and grammar have to be revised carefully. Terms like “in vivo” have to be written in italics (e.g. see l. 58). Use a uniform style for unit, such as “min” vs “minutes” vs “hours” and so on (e.g. see lines 86, 87, 107, 110, 125, 141, 143, 152, 156 etc.). Please use a uniform style for SI units (e.g. see “µl” vs “mL” in lines 155 vs 162 etc.). Use a uniform style for significant differences (e.g. compare l. 366 “p < 0.05” vs l. 359 “P < 0.01”. Check the whole text. Please use a uniform style for the term “figure” vs “Fig.” (instead of alternating between both forms; see lines 194, 197, 200, 257 etc.). Sometimes, thousands separators are missing in numbers (e.g. see l. 87: “6000”, l. 156 etc.). Define all used abbreviations when used for the first time in the text. After the term was defined, please uniformly use the abbreviation instead of alternating between full names and the abbreviation (e.g. see: "TLR4” in l. 67, “bw” in l. 108, 109 ,” PCR” l. 128, “TBST” l. 143, “SDS-PAGE, BCA, PVDF” lines 138-140, “CTAB/SDS” in l. 167 etc.). Sometimes space characters are missing or too much (e.g. see l. 155 “100μL” vs 162 “0.5 mL/min”, l. 226 “P <0.01,” vs l. 228 “P < 0.05” vs l. 247 “p < 0.001”, also see l. 160 “25mM” etc.). In line 158, the term “ACQUITY UPLC BEH 158 C18” should not be written in capital letters. In general, the text was written in a rather sloppy style. It is recommended to consult a native English speaker and to make use or editorial services offered by the journal.
Reviewer 2 Report
Authors Sheng et al. have submitted a manuscript for consideration detailing a study that investigates a novel phytochemical extract to ameliorate pathophysiological changes associated with NAFLD in the liver, adipose and gut microbiota and tissue. The preventative role of the thyme polyphenol extract (TPE) is evident and appears to be dose-dependent. Given the relatively small number of suitable therapeutic options for NAFLD and metabolic disorder that are available or emerging, the study is timely and appropriate.
The study is well designed and presented, and the authors' conclusions are consistent with the data and support their hypotheses. As an aside, the inclusion of both IHC and western blots for TLR4 are welcomed, as these antibodies are notorious for non-specific binding and showing positive signals in knockout mice. In general, I have few comments as I think this is a polished study.
My comments for the authors:
1- The changes in TLR4 expression and signaling are evident, but the authors have not investigated changes to infiltrating immune cells. Therefore the changes in the immune receptors and immune signaling pathways noted could be from either changes in the epithelium and/or resident immune cells, e.g. Kupffer cells in liver, macrophages in adipose or intestinal macrophages/innate lymphoid cells. I think the authors should detail changes to KCs (f4/80 stain/PCR), monocytes (ly6C stain/PCR), and crown structures in adipose which are discernible in the H&E panels from Fig 2. Moreover, this is suggested by the changes in circulating endotoxin that could be leading to decreased immune activation or infiltration.
Expression of chemokines like Ccl2, for instance, could also provide insight to the immune infiltration in these models.
This would give a more comprehensive look into the effect of TPE on HFD-induced changes.
2- The changes in plasma endotoxin activity are connected to maintenance of the barrier via Zo-1 and Occludin staining. Do the authors know if TPE can bind LPS, thus inhibiting its activity in circulation?
3- Do the authors know if starting TPE therapy after the onset of liver, adipose and gut dysfunction presents would lead to protection? The current study only shows concurrent TPE therapy during the entire HFD feeding period.
